# Effect of Talc in Mixtures with Fly Ash on Sintering Crystalline Phases and Porosity of Mullite-Cordierite Ceramics

Marta Valášková [1,*], Veronika Blahůšková [1], Alexandr Martaus [1], Soňa Študentová [2], Silvie Vallová [1,2] and Jonáš Tokarský [1,3]

1   Institute of Environmental Technology, CEET, VSB-Technical University of Ostrava, 17. listopadu 2172/15, 708 00 Ostrava, Czech Republic; veronika.blahuskova@vsb.cz (V.B.); alexandr.martaus@vsb.cz (A.M.); silvie.vallova@vsb.cz (S.V.); jonas.tokarsky@vsb.cz (J.T.)
2   Department of Chemistry, VSB-Technical University of Ostrava, 17. listopadu 2172/15, 708 00 Ostrava-Poruba, Czech Republic; sona.studentova@vsb.cz
3   Nanotechnology Centre, CEET, VSB-Technical University of Ostrava, 17. listopadu 2172/15, 708 00 Ostrava-Poruba, Czech Republic
*   Correspondence: marta.valaskova@vsb.cz; Tel.: +420-597-327-308

**Abstract:** The effect of talc in the two mixtures with the representative sample of fly ash (Class F) was investigated at sintering temperatures of 1000, 1100, and 1200 °C. X-ray diffraction, thermal DTA/TGA, and mercury intrusion porosimetry analyses were applied to characterize the mineral phase transformation of talc and fly ash in cordierite ceramic. The influence of iron oxide on talc transformation to Fe-enstatite was verified by the simulated molecular models and calculated XRD patterns and the assumption of Fe-cordierite crystallization was confirmed. The fly ash mixtures with 10 mass% of talc in comparison with 30 mass% of talc at 1000 °C and 1100 °C showed higher linear shrinkage and lower porosity. At a temperature of 1200 °C, sintering expansion and larger pores in mullite and cordierite ceramics also containing sapphirine and osumilite demonstrated that magnesium in FA and Tc structure did not react with the other constituents to form crystalline cordierite. The ceramics produced in the present work using fly ash and talc have similar properties to the commercial ceramics produced at sintering temperatures higher than 1250 °C.

**Keywords:** talc; thermal changes; mullite; cordierite; porosity; X-ray diffraction; structure analysis

## 1. Introduction

Fly ash (FA) is one of the most abundant waste materials produced as a by-product of coal combustion, and despite their processing in various industries, disposal remains problematic [1]. Many works have aimed at the possibility of the application of different types of waste mixed with fly ash as raw materials. In the production of traditional ceramics, there is an effort to design ceramic bodies with the maximum amount of fly ash in the initial mixture. Coal ash was investigated from the 1990s for stoneware manufacture. Fly ash is a by-product of coal combustion captured by mechanical and electrostatic separators from the fuel gases of the power plants. According to the ASTM standard, there are three types of FA: class F, class C, and class N, based on the main content of the constituent material [2,3]. The properties of FA depend on the coal source, the method of combustion of power plants, storage, etc. Proper use of FA requires information on its chemistry and phase composition [4]. The main oxide components of FA are $SiO_2$, $Al_2O_3$, $Fe_2O_3$, $CaO$, $MgO$, $Na_2O$, and $K_2O$, and the crystalline phases are mullite ($3Al_2O_3 \cdot 2\,SiO_2$) and quartz ($SiO_2$) [5]. The ash fusion system corresponds to the pseudo-ternary system $Al_2O_3$–$SiO_2$–Base ($FeO + CaO + K_2O$). Huffman et al. [6] characterized ash fusion by a number of tests and temperatures of change to the formation of the liquid phase and phases identified on ternary phase diagrams. The partial ash melting at 400 °C and at temperatures from 900 to 1200 °C under reducing conditions was controlled by the phases occurring in the

FeO–Al$_2$O$_3$–SiO$_2$ phase diagram. Under oxidizing conditions up to 1200 °C, the amount of potassium-bearing minerals influenced the amount of the glassy phase. Above 1200 °C, Ca and Fe were effective fluxing elements. Shrinkage of ash during heating is associated with the fusion of particles and the reduction of porosity [7]. The processes associated with ash fusion as temperature rises involve melting and reactions to form the first melt (liquid) phase. The most significant phase of the melt is rich in iron, and often has a composition close to iron cordierite. The lowest temperature at which the ash pellets were prepared was approximately 1200 °C.

Fly ash containing a large amount of reactive SiO$_2$ and alumina Al$_2$O$_3$ is very suitable to produce mullite (3Al$_2$O$_3$·2SiO$_2$) [8] and is an alternative source of minerals with a composition close to building ceramics [9]. Currently, attention is being given to ceramic industry applications in the development of porous ceramic microfiltration membranes [10,11], ceramic tiles sintered at a relatively low sintering temperatures [12,13], geopolymer products [14], etc.

Raw talc is used in the ceramic materials industry either as a basic raw material or as a filler due to its chemical inertia and thermal stability as a non-plastic ceramic material in the production of wall and floor tiles, and porcelain prevents the cracking effect on the glazes [15]. Talc stone is a rock containing talc together with other minerals (chlorite, tremolite, and carbonates) and admixtures of these minerals are used to improve the properties of ceramics [16].

Talc, Mg$_3$Si$_4$O$_{10}$(OH)$_2$, is a hydrated trioctahedral 2:1 layer silicate of the talc-pyrophyllite group. The 2:1 layer is composed of two opposing tetrahedral sheets with the octahedral sheet between. The apical oxygens form a plane of anions common to adjacent tetrahedral and octahedral sheets. The OH group is at the center of each six-fold ring of tetrahedra. The 2:1 layers are electrostatically neutral. The primary atomic forces holding the 2:1 layers together are van der Waals bonds [17].

Natural raw clay materials containing oxides MgO, Al$_2$O$_3$, and SiO$_2$ are used for the synthesis of cordierite and steatite ceramics. Cordierite, Mg$_2$Al$_4$Si$_5$O$_{18}$, and steatite, MgSiO$_3$ (occurring in four polymorphic forms as enstatite, protoenstatite, clinoenstatite, and high temperature clinoenstatite [18]), are the major components of the MgO–Al$_2$O$_3$–SiO$_2$ ternary system [19]. Pure cordierites were sintered from talc, FA, silica, and alumina [20]. Steatite and cordierite were prepared from natural raw materials, mainly with kaolinite and talc [21], or talc and montmorillonite [22]. Mixtures of clay minerals suitable for the preparation of cordierite ceramics are summarized in [23]. Mixtures of kaolin stoneware clay and high temperature FA were developed for the dry pressed ceramic tiles prepared by single-sintering technology [24]. In the case of FA–talc mixtures, the effect of the addition of 0–100 mass% of talc was studied on the sintering characteristics of FA-based ceramic tiles [25]. Clay minerals in FA–clay mixtures may strongly affect the properties of the manufactured ceramics due to their dehydration at lower temperatures and phase changes during high-temperature reactions [26,27].

FA, as a typical solid waste produced during power plant operation, is used in commercial ceramics obtained at sintering temperatures of 1250–1300 °C. In this work, FA, as the main component in mixtures with talc (amount of 10 and 25 mass %), was used in a green synthesis route to prepare the FA–talc based ceramics obtained after heating at low sintering temperatures of 1000, 1100, and 1200 °C. This work mainly aimed to investigate the effect of the talc on the evolution of mineral phases and the pore characteristics in the mullite-cordierite ceramic body. Finally, the phase composition and porosity of mullite and cordierite ceramics prepared by an efficient green route of FA recycling using a small amount of talc at lower sintering temperatures are compared and discussed.

## 2. Materials and Methods

### 2.1. Materials and Sample Preparation

The FA sample from the combustion of black coal was obtained from an electrostatic precipitator in a power station (Czech Republic). The talc (Tc) sample was supplied by

KOLTEX COLOR, the ceramic type KT5, s.r.o. (Czech Republic). The FA:Tc mixtures of their % mass ratio of 90:10 ($FA_{90}Tc_{10}$ sample) and 75:25 ($FA_{75}Tc_{25}$ sample) were homogenized in bottles at 40 rpm for one hour (Heidolph Reax overhead shaker, REAX 20/4, Merck company). Then, the samples (each of 100 g) were milled at 300 rpm for 15 min in agate planetary mill FRITSCH-Pulverisette 6 using 20 balls with a diameter of 10 mm. The $FA_{90}Tc_{10}$ and $FA_{75}Tc_{25}$ ceramic mixtures were prepared into a slurry with distilled water (about 20% by weigh), manually pressed into cubic cavities of the metal molds of 8 $cm^3$ dimension, left for 24 hours at room temperature, and dried at 110 °C for five hours in an oven. Dry $FA_{90}Tc_{10}$ and $FA_{75}Tc_{25}$ samples were taken out of the molds and sintered at temperatures of 1000, 1100, and 1200 °C in order to prepare the ceramic samples $FA_{90}Tc_{10}$-1000, $FA_{90}Tc_{10}$-1100, $FA_{90}Tc_{10}$-1200, and $FA_{75}Tc_{25}$-1000, $FA_{75}Tc_{25}$-1100, $FA_{75}Tc_{25}$-1200. Sintering occurred in an electrical laboratory furnace LH15/13 with the heating ramp of 12 °C/min to the desired temperature (1000, 1100, and 1200 °C) maintained for two hours.

### 2.2. Methods

Elemental analysis of samples determined using a SPECTRO XEPOS energy dispersive X-ray fluorescence (ED-XRF) spectrometer (Spectro Analytical Instruments, Kleve, Germany) was recalculated to the stoichiometric metal oxide concentrations.

The X-ray powder diffraction (XRD) patterns were recorded under $CoK_\alpha$ radiation ($\alpha_1$ = 0.1789 nm and $\alpha_2$ = 0.1793 nm) at 40 kV and 40 mA on the Rigaku SmartLab diffractometer (Rigaku Corporation, Tokyo, Japan) equipped with the D/teX Ultra 250 detector (Rigaku Corporation, Tokyo, Japan). Diffraction took place in reflection mode on the samples on the Si-holder at the rotation speed of 15 rpm $min^{-1}$ in symmetrical Bragg–Brentano diffraction geometry from the 5 to 80° $2\theta$ with a step size of 0.01° and speed of 0.5 deg $min^{-1}$.

The thermal decomposition of talc and mullite performed on the Simultaneous DTA/TGA SDT 650 system (TA Instruments, New Castle, USA) occurred in an air dynamic atmosphere with a flow rate of 0.1 l $min^{-1}$ at a heating rate of 10 °C $min^{-1}$ over the range 25–1200 °C.

The porosity data of the ceramic samples were obtained using a mercury intrusion porosimeter AutoPore IV 9500 (Micromeritics Instrument Corporation, Norcross, GA, USA).

The measurement of surface area (SA) was performed with a Sorptomatic 1990 (Thermo Electron Corporation, Waltham, MA, USA) at liquid-nitrogen temperature. Nitrogen gas was used as an adsorbate.

The Biovia Materials Studio modeling environment (MS) allows for models of the talc and enstatite crystal structures to be built. Charge equilibration method [28] and universal force field [29] was used to calculate the atomic charges and to parameterize the atoms in all models, respectively. Geometry optimization of the prepared models was performed under periodic boundary conditions in the MS/Forcite module using the Smart algorithm with $5 \cdot 10^4$ iteration steps. Convergence criteria used for energy and force were 0.001 kcal/mol and 0.05 kcal/mol/nm, respectively. Basal distances in the optimized models of the talc interlayer space were obtained from the X-ray diffraction patterns simulated in the MS/Reflex module under conditions similar to the experimental measurements.

### 3. Results

*3.1. Talc (Tc) and Fly Ash (FA)*

3.1.1. Structure and Chemistry

The chemical analysis of Tc sample revealed less than 2% CaO and 1% $Fe_2O_3$, except for the Mg and Si elements structurally bonded in talc (Table 1). The crystalline phases in the XRD pattern of Tc (Figure 1) are talc (PDF No. 01-080-1119, $Mg_3Si_4O_{10}(OH)_2$) and the accessory minerals clinochlore (PDF No. 01-074-1036, $Mg_{4.54}Al_{0.97}Fe_{0.46}Mn_{0.03}(Si_{2.85}Al_{1.15}O_{10})(OH)_8$) and dolomite (PDF No. 01-075-6391, $CaMg(CO_3)_2$).

**Table 1.** Chemical compositions of metal oxides (mass %) and surface area (SA, m$^2$/g) of the fly ash (FA) and talc (Tc) samples.

| Sample | SiO$_2$ | TiO$_2$ | Al$_2$O$_3$ | Fe$_2$O$_3$ | CaO | MgO | K$_2$O | Na$_2$O | SO$_3$ | P$_2$O$_5$ | L.O.I. [1] | SA |
|---|---|---|---|---|---|---|---|---|---|---|---|---|
| FA | 52.66 | 1.13 | 25.6 | 6.87 | 2.5 | 2.57 | 3.14 | 1.65 | 0.45 | 0.69 | 2.61 | 1.4 |
| Tc | 56.47 | 0.01 | 0.83 | 0.37 | 1.95 | 32.74 | 0 | 0 | 0.03 | 0.07 | 7.43 | 6 |

[1] L.O.I. (loss on ignition).

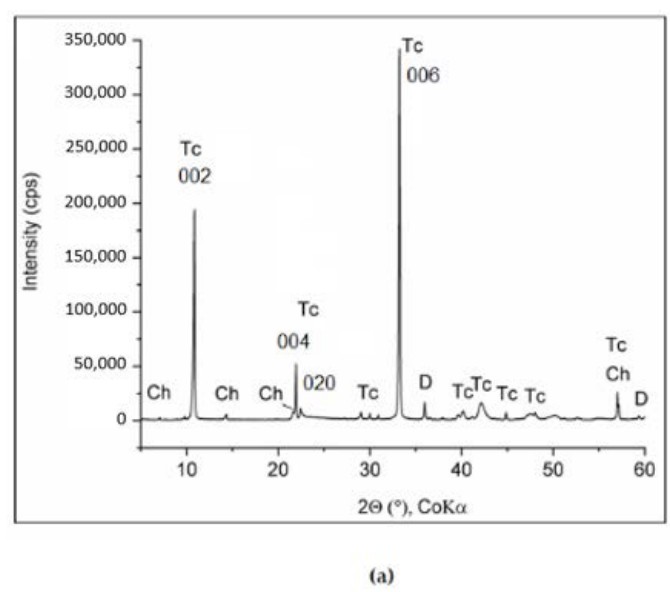

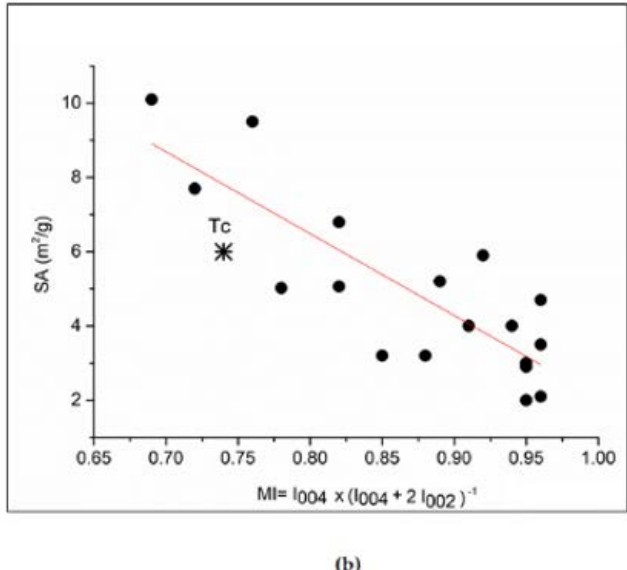

(a)　　　　　　　　　　　　　　　(b)

**Figure 1.** X-ray diffraction (XRD) pattern of the talc (Tc) sample (Tc-talc, Ch-clinochlore, D-dolomite) (**a**). Surface area as a function of the morphology index (MI) taken from [30] and implemented by the Tc sample (**b**).

The basal planes (00l) in the 2:1 type layer structure of Tc were preferably oriented with the tendency to lie parallel to the plane. The XRD basal peaks of 002, 004, and 006 of Tc in the sample pressed in the rotation holder were intense, while the intensity of the non-basal diffractions was suppressed (Figure 1a).

The intensities of the preferred orientation on crystallites lying within the surface can be obtained to calculate the "ideal" powder patterns from known atomic positions in the crystal structure of talc, corresponding to the formula Mg$_3$Si$_4$O$_{10}$(OH)$_2$ (Figure 2).

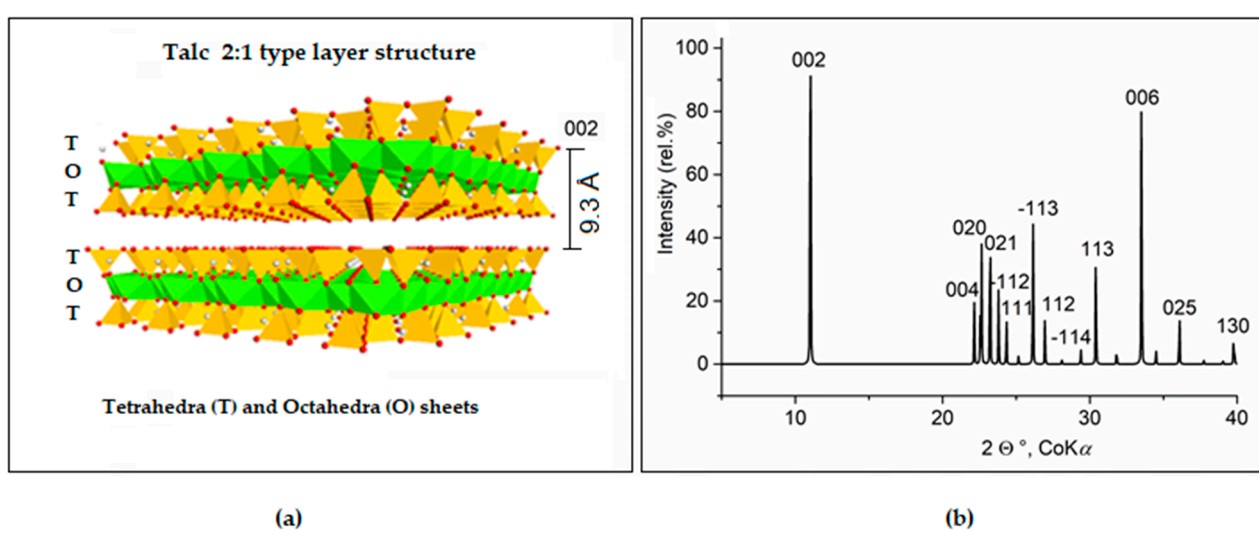

(a)　　　　　　　　　　　　　　　(b)

**Figure 2.** The molecular model of talc (**a**) and simulated XRD pattern [31] (**b**).

The initial Tc model (Figure 2a) with the interlayer value $d_{002}$ = 9.3 Å observed on the measured XRD pattern (Figure 1a) was built with the unit cell parameters $a$ = 5.28 Å, $b$ = 9.15 Å, $c$ = 18.92 Å, $\alpha = \gamma = 90°$, and $\beta = 100°$ [31]. The XRD pattern calculated for the structure of the Tc model (Figure 2b) showed diffractions and their relative intensity distribution ($I_{rel}$) from the basal planes (e.g., 002, 004, 006) and prism planes (e.g., 02$l$, 11$l$).

In the literature, the sensitivity of the intensity of basal diffractions to the Tc particles' morphology was studied on the Tc samples selected from different localities [30]. Studies have revealed that the intensities of the 004 and 020 diffractions are the most sensitive to the particles' morphology, and calculated the XRD morphology index (MI) according to Equation (1):

$$MI = I_{004} \, (I_{004} + 2I_{020})^{-1} \tag{1}$$

The texture of Tc classified according to the MI were declared as blocky (MI = 0), partially blocky (MI = 0.71), and platy (MI = 1).

The data published about surface area (SA) and MI [30] allowed us to compute the relationship between MI and SA (Figure 1b), which can be described by a linear regression function in Equation (2):

$$SA = 24.13 - (22.05 \cdot Mi) \tag{2}$$

For the Tc sample, MI = 0.74 calculated according to Equation (1) corresponds to a partially blocky particle morphology with a SA = 7.8 m$^2$ calculated according to Equation (2) in comparison with the measured SA = 6.0 m$^2$ (Table 1).

The XRD pattern of the FA sample (Figure 3) contained peaks of crystalline mullite (PDF No. 01-079-1456), quartz (PDF No. 01-086-1628), and $Fe_3O_4$ (PDF No. 00-019-0629) over the broad halo peak (20–40 $2\theta°$).

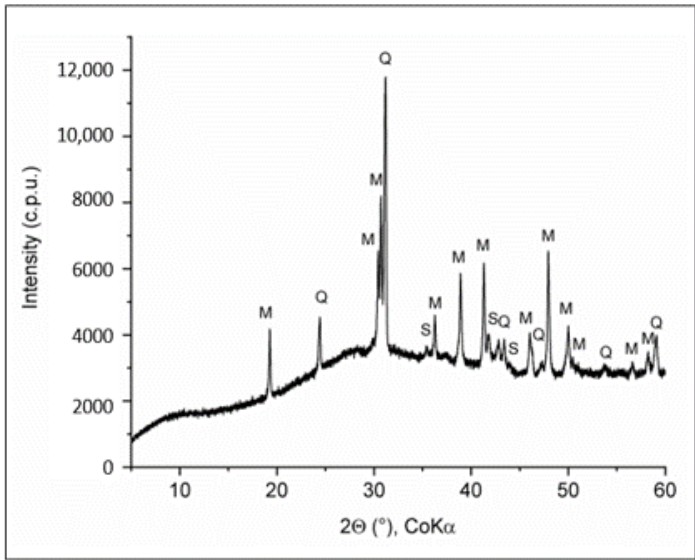

**Figure 3.** XRD pattern of fly ash (FA). M—mullite, Q—quartz, S—spinel, magnetite, $Fe_3O_4$.

Chemical XRF analysis of FA (Table 1) primarily indicated $SiO_2$ and $Al_2O_3$ aside from $Fe_2O_3$ (6.87 mass%). Alkaline-earth and alkali metals (CaO 2.50 mass%, $K_2O$ 3.14 mass% and $Na_2O$ 1.65 mass%) were not observed in the crystalline phases (Figure 3) and their occurrence in the X-ray undetectable phases in the amorphous halo from about 20 to 45 ($2\theta$) over the XRD background can be assumed. Chemical analysis of a representative FA sample (Table 1) was comparable with analyses of other fly ashes containing CaO less than 7% by mass (e.g., [9,11]) and met the requirements of FA Class F, according to the ASTM classification [2].

### 3.1.2. Thermal Transformation

The thermal changes of the Tc and FA samples and their mixtures $FA_{90}Tc_{10}$ and $FA_{75}Tc_{25}$ (Figure 4) were studied on the TG (Figure 4a) and DTA curves (Figure 4b) with a temperature increasing up to 1200 °C. Tc started to dehydroxylate at about 800 °C, resulting in a 5.7% mass loss in the range of 25–800 °C and about a 4% mass loss in the range of 800–1200 °C. The endothermic peaks at 936.5 and 1045 °C were attributed to the decomposition of Tc to amorphous magnesium silicate (enstatite), amorphous silica, and water vapor [20], according to the reaction in Equation (3):

$$Mg_3Si_4O_{10}(OH)_2 \rightarrow 3(MgSiO_3) + SiO_2 + H_2O \tag{3}$$

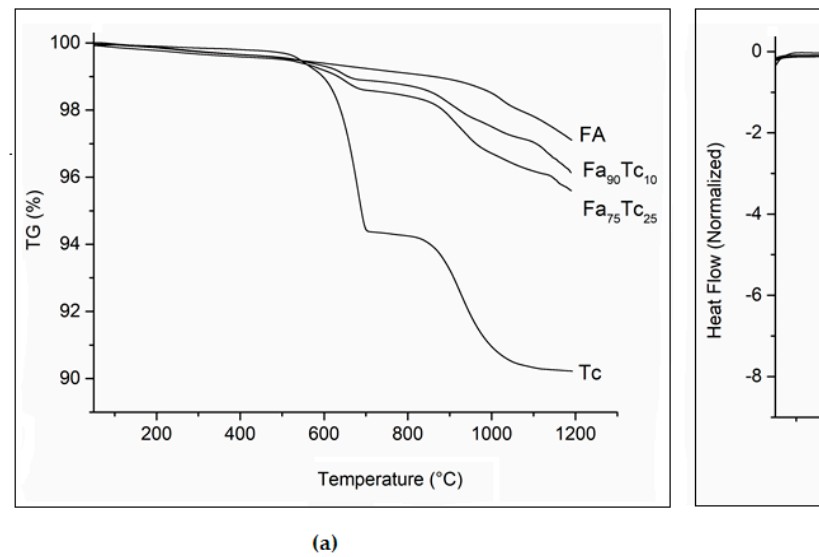 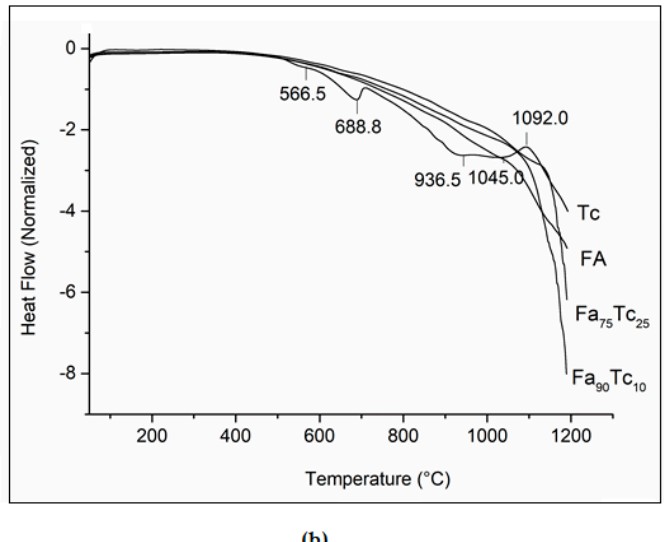

(a)  (b)

**Figure 4.** The thermal decomposition of FA and Tc on the TG (**a**) and DTA (**b**) curves.

The exothermic maximum at 1092 °C corresponds to the crystallization of enstatite [32].

Dehydroxylation of clinochlore in Tc (see XRD pattern in Figure 1a) occurred at about 600 °C (Figure 4b) to the spinel ($MgAl_2O_4$), forsterite ($MgSiO_4$), enstatite $MgSiO_3$ and water, according to the reaction in Equation (4) [33]:

$$Mg_5Al_2Si_3O_{10}(OH)_8 \rightarrow MgAl_2O_4 + Mg_2SiO_4 + 2MgSiO_3 + 4H_2O \tag{4}$$

An endothermic maximum at 688.8 °C corresponded to the decomposition of dolomite to lime (CaO), periclase (MgO), and carbon dioxide [32], according to the reaction in Equation (5):

$$CaMg(CO_3)_2 \rightarrow CaO + MgO + 2CO_2 \tag{5}$$

The total mass loss in TG from 25 to 1200 °C was for FA 2.9%, Tc 9.7%, and for both $FA_{90}Tc_{10}$ and $FA_{75}Tc_{25}$ mixtures of 6.0% and 5.9%, respectively.

### 3.2. Talc-Fly Ash Ceramics Prepared at 1000, 1100, and 1200 °C

3.2.1. Mineral Phases in Ceramic Samples

The XRD patterns of the $FA_{90}Tc_{10}$ and $FA_{75}Tc_{25}$ ceramic samples sintered at temperatures of 1000, 1100 and 1200 °C (Figure 5) showed mineral phases identified using the PDF 2 database as follows:

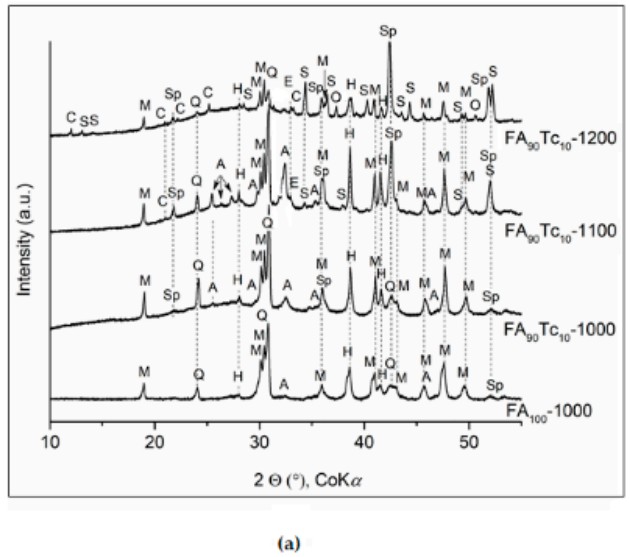
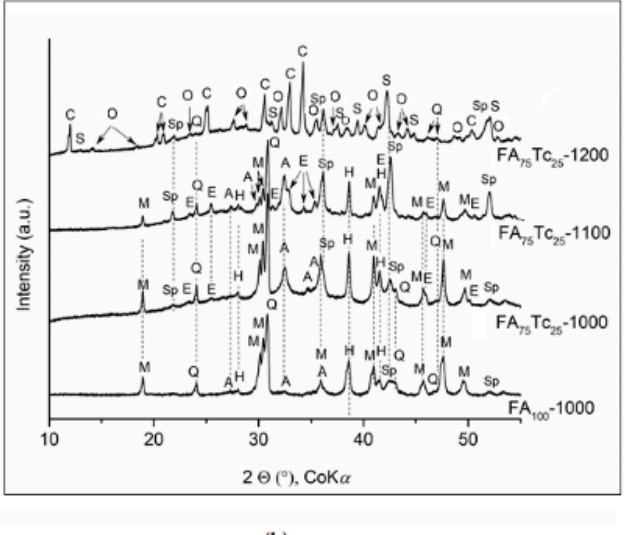

**Figure 5.** XRD patterns of ceramic samples $FA_{90}Tc_{10}$ (**a**) and $FA_{75}Tc_{25}$ (**b**) sintered at 1000, 1100, and 1200 °C. A—anorthite, C—cordierite, Q—quartz, E—enstatite, H—hematite, M—mullite, O—osumilite, S—sapphirine, Sp—spinel.

Anorthite (PDF No. 01-071-0748, $(Na_{0.45}Ca_{0.55})(Al_{1.55}Si_{2.45}O_8)$; cordierite (PDF No. 01-012-0303, $Mg_2Al_4Si_5O_{18}$); Fe-cordierite (PDF No. 00-002-0646, $(Mg_{0.57}Fe_{0.43})_2 Al_4Si_5O_{18}$); quartz (PDF No. 01-075-8320, $SiO_2$); enstatite (PDF No. 01-084-2026, $Mg_{1.79}Fe_{0.21}Si_2O_6$); hematite (PDF No. 01-079-0007, $Fe_2O_3$); mullite (PDF No. 01-079-1456, $Al_{4.54}Si_{1.46}O_{9.73}$); osumilite (PDF No. 01-084-1719, $(K_{0.90})(Mg_{1.68}Fe_{0.32})(Al_{2.9} Fe_{0.09})(Si_{10.2}Al_{1.8})O_{30}$); sapphirine (PDF No. 01-076-0537, $(Mg_{3.78}Al_{4.22})(Si_{1.91}Al_{4.09})O_{20}$), and spinel (PDF No. 01-021-0540, $Mg(Al,Fe)_2O_4$).

Chemical analysis of the FA sample (Table 1) also contained, in addition to the elements Al and Si bound in mullite and quartz, Fe (bound in magnetite), Ca (bound in dolomite and Ca and Na in anorthite), and K from sedimentary rocks (Table 2).

**Table 2.** Crystalline phases identified in the X-ray diffraction (XRD) patterns of ceramic samples.

| Phase | $FA_{90}Tc_{10}$ | | | $FA_{75}Tc_{25}$ | | |
|---|---|---|---|---|---|---|
| | 1000 | 1100 | 1200 | 1000 | 1100 | 1200 |
| Mullite | ++ | ++ | ++ | ++ | + | 0 |
| Quartz | + | + | + | + | + | + |
| Spinel | + | + | ++ | + | ++ | + |
| Hematite | + | + | + | + | + | 0 |
| Anorthite | + | + | 0 | + | + | 0 |
| Enstatite | 0 | + | 0 | + | + | 0 |
| Sapphirine | 0 | 0 | + | 0 | 0 | ++ |
| Cordierite | 0 | 0 | + | 0 | 0 | ++ |
| Osumilite | 0 | 0 | + | 0 | 0 | + |

Mullite is the stable crystalline phase of the binary system $Al_2O_3$–$SiO_2$ at atmospheric pressure above a temperature of 700 °C. In the literature, mullite reaction products in water vapor to alumina ($\alpha$-$Al_2O_3$) and silica [8], mullite melting incongruently under equilibrium conditions in the presence of $\alpha$-$Al_2O_3$ [34], the recession of mullite ceramics at the silica-rich glass grain phase at about 1050 °C, and decomposition of mullite to alumina at about 1250 °C were documented [35].

The set of our samples showed similar results with the laboratory experiments on the high-temperature behavior of coal ash in reducing and oxidizing atmospheres [6]. The $FA_{90}Tc_{10}$-1000 and $FA_{75}Tc_{25}$-1000 samples in comparison with FA-1000 (Figure 5) contained newly generated mullite, quartz, Mg–Al–Fe spinel, increased peaks of hematite, and

unchanged anorthite. In the $FA_{75}Tc_{25}$-1000 sample, chlorite, a layer-silicate mineral rich in iron and magnesium, occurred in the Tc sample (Figure 1a), which can transform to glassy or poorly crystalline phases when iron oxides react with quartz and talc is decomposed to enstatite (Figure 5b).

At a temperature of 1100 °C, anorthite and spinel increased in both of the $FA_{90}Tc_{10}$-1100 and $FA_{75}Tc_{25}$-1100 samples, evidencing phase changes from high-melting mullite in the presence of enstatite. Interaction of the components Fe in FA to the cordierite substrate at a temperature above 1000 °C [36] and the migration of Fe to cordierite substrate pores and substitution reaction of Mg at 900 and 1100 °C [37] can be assumed.

At a temperature of 1200 °C, the $FA_{90}Tc_{10}$-1200 sample contained the main phases of mullite, quartz, spinel, and the minor phases as cordierite, sapphirine, and osumilite. The $FA_{75}Tc_{25}$-1200 main phases were cordierite and sapphirine, and the minor phases were osumilite, spinel and quartz. The mineral phases in both samples correspond with the cordierite-sapphirine- spinel glass-ceramic system prepared at 1050 °C [38].

The Tc sample heated at 1100 °C to the enstatite Tc-1100 sample was prepared based on the DTA exothermal maximum at 1092 °C (Figure 4b). The XRD pattern of the Tc-1100 sample (Figure 6) showed a weakly crystalline phase assigned to the orthoenstatite, (PDF No. 01-082-3780, $MgSiO_3$, $a$ = 18.158 Å, $b$ = 8.780 Å, $c$ = 5.157 Å, $\alpha = \beta = \gamma = 90°$). Enstatite in the $FA_{75}Tc_{25}$-1100 ceramic sample (Figure 5b) was assigned to the Mg–Fe orthoenstatite, (PDF No. 01-084-2026, $Mg_{1.79}Fe_{0.21}Si_2O_6$, $a$ = 18.242 Å, $b$ = 8.803 Å, $c$ = 5.198 Å, $\alpha = \beta = \gamma = 90°$). Due to the similarity in the ionic radii of $Mg^{2+}$ and $Fe^{2+}$, it can be assumed that Mg–Fe silicate solid solutions should show thermodynamic behavior. The presumption of the presence of Mg–enstatite or Mg,Fe–enstatite can be verified by modeling the crystal structures and their XRD patterns.

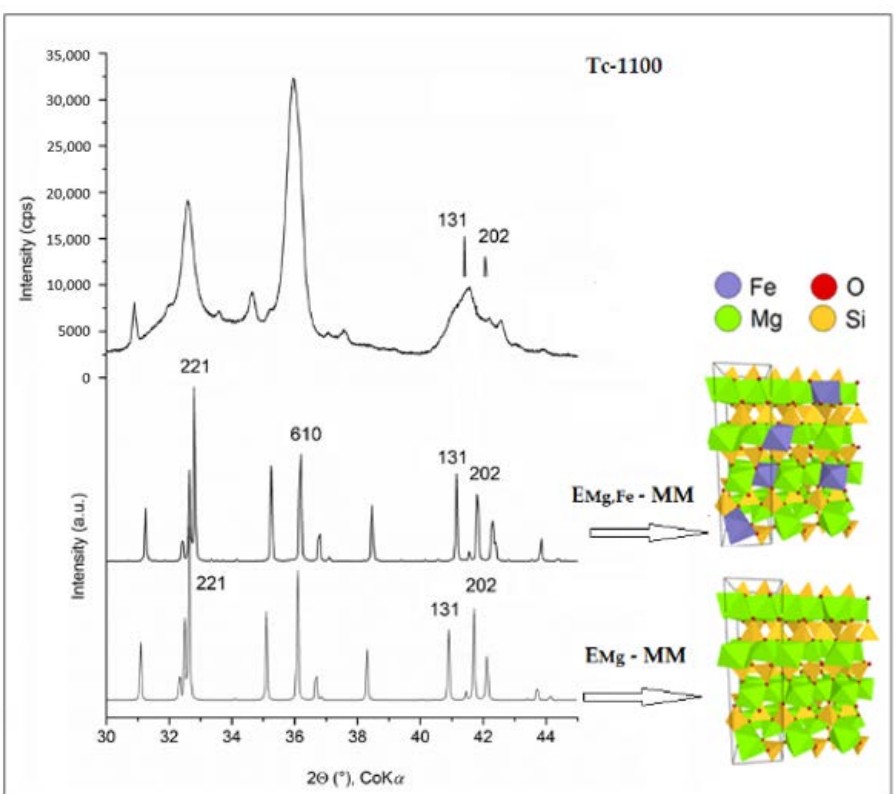

**Figure 6.** XRD patterns of the enstatite Tc-1100 sample in comparison with the calculated XRD patterns of the structure models of enstatites $E_{Mg}$–MM and $E_{Mg,Fe}$–MM.

Molecular models (MM) were built based on the enstatite unit cells [31] with a crystallochemical formula $Mg_{12}Si_{16}O_{40}(OH)_8$ and $Mg_{16}Si_{16}O_{48}$, respectively. The initial model of the enstatite ($E_{Mg}$–MM) (unit cell parameters $a$ = 18.31 Å, $b$ = 8.93 Å, $c$ = 5.23 Å,

$\alpha = \beta = \gamma = 90°$) was also prepared in an iron-substituted form ($E_{Mg,Fe}$–MM) so that its composition corresponded to the formula $(Mg_{1.79}Fe_{0.21})Si_2O_6$ in PDF No. 01-084-2026. In light of the fact that the Mg and Fe content in this case led to a unit cell having the formula $(Mg_{14.32}Fe_{1.68})Si_{16}O_{48}$, a superstructure containing three unit cells with the formula $(Mg_{42.96}Fe_{5.04})Si_{48}O_{144}$ (approximated as $(Mg_{43}Fe_5)Si_{48}O_{144}$) was prepared.

The calculated XRD patterns of the enstatite models $E_{Mg}$–MM and $E_{Mg,Fe}$–MM slightly varied in the 221 peak position and intensity ratio between peaks 131 and 202 due to the Fe substitution (Figure 6). Figures show evidence that the structural model due to the overlapping diffraction peaks 131 and 202 in the poorly crystalline enstatite sample, similar to the Mg-Fe enstatite in the FA$_{75}$Tc$_{25}$-1100 sample, cannot be unambiguously assigned. However, the overall similarity of the calculated XRD patterns of the $E_{Mg}$–MM and $E_{Mg,Fe}$–MM structures does not directly deny the possibility of the presence of a Mg–Fe enstatite phase in the sample.

### 3.2.2. Linear Shrinkage and Porosity

In the literature, the sintering process is divided into shrinking (the initial stage) and phase formation and crystal growth (the intermediate and final stages) (e.g., [12]). The shrinkage at 1000 °C was attributed to the development of a low viscosity liquid phase, which can cause dimensional deformation of the samples.

The linear shrinkage (L) of the sintered samples is generally calculated according to Equation (6):

$$(L)~(\%) = [(L_g - L_s)/L_g] \times 100 \tag{6}$$

where $L_g$ and $L_s$ are the side lengths (mm) of the green (g) and sintered samples (s), respectively.

At sintering temperatures, all mixtures containing CaO bound in rich crystalline phases showed the expansion (Figure 7). For our samples, the linear shrinkage value was the average of three replicate measurements with a maximum standard deviation of 0.04%.

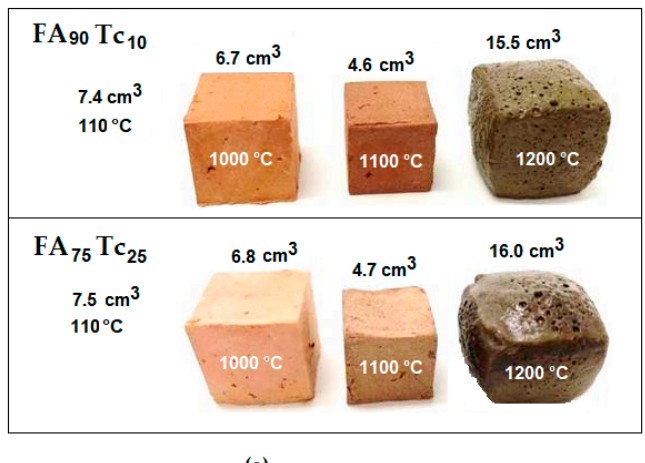

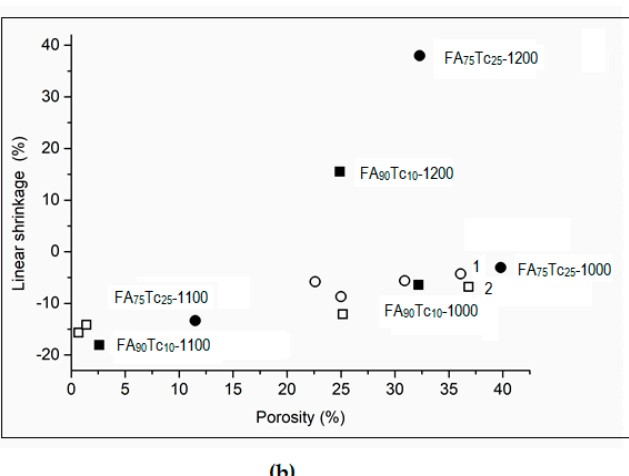

(a)  (b)

**Figure 7.** Photographs of the ceramic cubes sintered at different temperatures (**a**). Relation between the linear shrinkage and apparent porosity (**b**) in the studied samples in comparison with the literature: 1 [39], 2 [12].

Linear shrinkage in the FA$_{90}$Tc$_{10}$ sample dried at 110 °C increased at the sintering temperatures 1000 °C and 1100 °C to about 6.45 ± 0.01 and 18.08 ± 0.01%, respectively, but only to about 3.08 ± 0.01 and 13.33 ± 0.01% in the FA$_{75}$Tc$_{25}$ sample. Sintering expansion occurring at 1200 °C was determined to be 15.50 ± 0.04% in FA$_{90}$Tc$_{10}$-1200 and 37.97 ± 0.04% in the FA$_{75}$Tc$_{25}$-1000 samples (Table 3, Figure 7).

**Table 3.** Linear shrinkage and selected mercury porosimetry data.

| Properties | Samples: -1000 | $FA_{90}Tc_{10}$- | | $FA_{75}Tc_{25}$- | | |
|---|---|---|---|---|---|---|
| | | -1100 | -1200 | -1000 | -1100 | -1200 |
| Linear shrinkage (%) | 6.45 | 18.08 | +15.50 | 3.08 | 13.33 | +37.97 |
| Porosity (%) | 32.2 | 2.6 | 24.9 | 39.8 | 11.5 | 32.3 |
| Average pores diameter (µm) | 0.49 | 0.08 | 0.02 | 0.49 | 0.66 | 0.01 |
| Total pores area $m^2$/g | 1.66 | 0.62 | 41.6 | 2.1 | 0.34 | 76.3 |

The behavior of the shrinkage at the temperatures of 1000, 1100, and 1200 °C can also be explained by the formation of crystalline phases during sintering. Dehydroxylation of talc resulted in enstatite, silica, and water (Equation (3)) and mullite containing the silica-rich glass grain at about 1050 °C. This silica dissolves to a liquid phase at higher temperatures. Sintering started at lower temperatures for the samples with glass, whereas a second shrinkage zone was observed for temperatures approaching 1100 °C. The content of CaO is one of the main factors determining the mineralogy of the sintered body and the extent of glass devitrification [40]. The reaction of CaO with the silica led to the formation of aluminosilicates and limited formation of a liquid phase during sintering, thereby resulting in a smaller shrinkage. At a temperature of 1200 °C, mullite appeared in $FA_{90}Tc_{10}$-1200 and cordierite in $FA_{75}Tc_{25}$-1200, and also sapphirine $(Mg_{3.78}Al_{4.22})(Si_{1.91}Al_{4.09})O_{20}$ as well as osumilite $(K_{0.90})(Mg_{1.68}Fe_{0.32})(Al_{2.9} Fe_{0.09})(Si_{10.2}Al_{1.8})O_{30}$ in both of the samples (Figure 5, Table 2). The result demonstrated that Mg practically did not react with the other constituents to form crystalline phases (cordierite). The moisture expansion at 1200 °C can be related to the amount of CaO and decomposition of talc to amorphous magnesium silicate (enstatite), amorphous silica, and water vapor (Equation (3)). Linear shrinkage and moisture expansion of our samples and those reported in a literature [12,39] correlated well with the porosity (Figure 7b).

The total pore area (Table 3) of the $FA_{90}Tc_{10}$ and $FA_{75}Tc_{25}$ ceramic samples formed during sintering at 1000, 1100, and 1200 °C revealed the characteristic pore size distribution (Figure 8). Cumulative pore volumes for the $FA_{90}Tc_{10}$ ceramic samples (Figure 8a) were lower in comparison to the $FA_{75}Tc_{25}$ ceramic samples (Figure 8b). The first derivate showed a well–defined peak of pore radii centered at 1.12 µm for $FA_{90}Tc_{10}$-1000 (Figure 8c) and at 0.9 µm for $FA_{75}Tc_{25}$-1000 (Figure 8d). The pore diameter observed for $FA_{90}Tc_{10}$-1200 and $FA_{75}Tc_{25}$-1200 had a largely different threshold observed on the three peaks maxima distribution. Cumulative pore volume for the $FA_{90}Tc_{10}$-1100 sample was negligible, and only a slightly higher cumulative pore volume was observed for the $FA_{75}Tc_{25}$-1100 sample.

Positions of the $FA_{90}Tc_{10}$-1200 and $FA_{75}Tc_{25}$-1200 samples in the $MgO–Al_2O_3–SiO_2$ phase diagram are shown in Figure 9a. Porosity analysis results are provided in Figure 9b.

Both $FA_{90}Tc_{10}$-1200 mullite ceramics and$FA_{75}Tc_{25}$-1200 cordierite ceramics contained three phases, sorted in descending order according to the quantity: mullite > spinel > cordierite and cordierite > mullite > spinel, respectively. Based on the chemistry, the samples were plotted among the phases in the $MgO–Al_2O_3–SiO_2$ ternary system (Figure 9a).

Porosity analysis (Figure 9b) revealed a significant increase in average pore size and pore distribution at 1200 °C, which may be due to the rapid growth of the particles and the penetration of a small amount of the liquid glass phase into several small gaps between the particles [10]. Similar to previously published data [10], open porosity decreased strongly from 1000 to 1100 °C, and increased intensively from 1100 to 1200 °C (Figure 9b).

Compared to the $FA_{90}Tc_{10}$-1200 sample, the $FA_{75}Tc_{25}$-1200 sample had a larger content of pores (Figure 8c,d), creating nearly twice as large a total pore area (Table 3). Differences between these two ceramics were observed on the average pore diameters at 1100 °C, strongly reduced in the $FA_{90}Tc_{10}$-1100 sample, and the smaller total pore area and low porosity (slightly higher in the $FA_{75}Tc_{25}$-1200 sample) (Figure 9b). The difference in pore diameter was not observed at 1200 °C while at a similar porosity, the high total pore area was measured for the $FA_{75}Tc_{25}$-1200 sample.

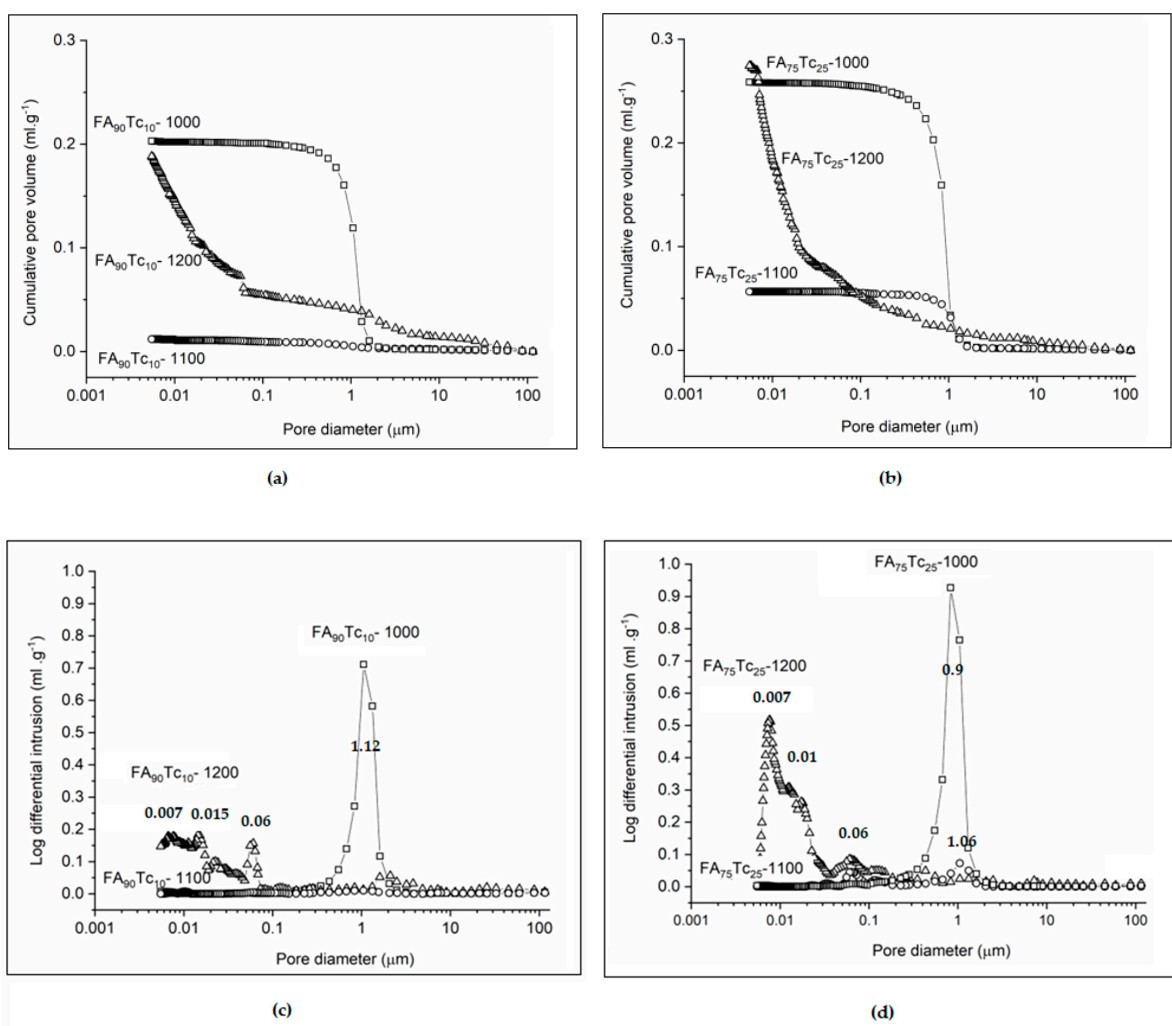

**Figure 8.** Cumulative intrusion in the ceramic samples: $FA_{90}Tc_{10}$ (**a**) and $FA_{75}Tc_{25}$ (**b**) and the derivate intrusion for $FA_{90}Tc_{10}$ (**c**) and $FA_{75}Tc_{25}$ (**d**).

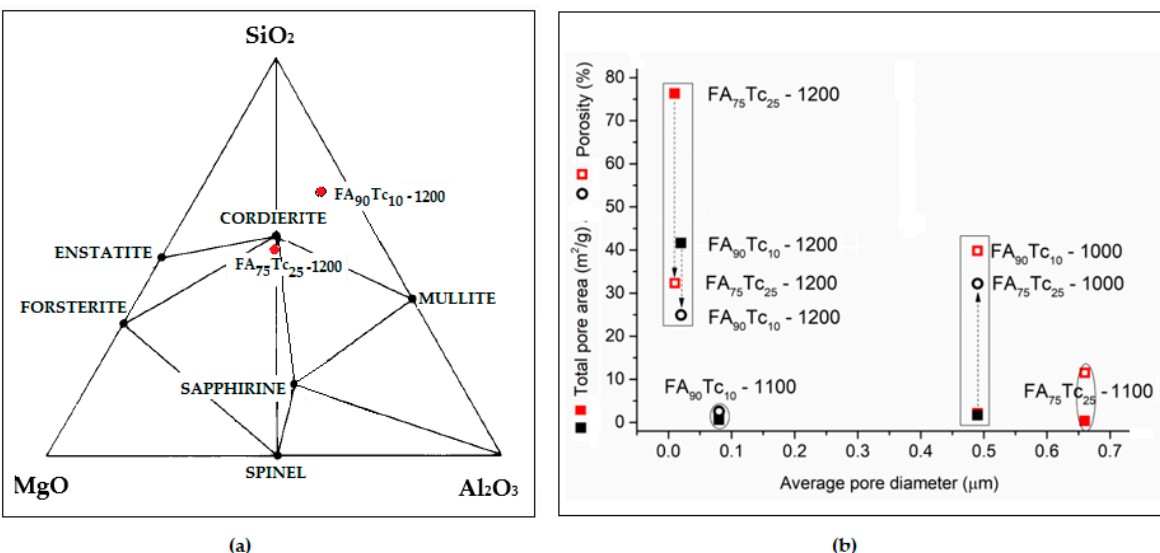

**Figure 9.** Phase distribution in the $MgO–Al_2O_3–SiO_2$ system showing fields of crystalline phases, tie lines [18], and positions of the $FA_{90}Tc_{10}$-1200 and $FA_{75}Tc_{25}$-1200 samples (**a**). Porosity and total pore areas in relation to the pore diameter in the sintered ceramic samples (**b**).

## 4. Conclusions

Two green ceramic materials containing talc (10 mass% and 25 mass%) in fly ash mixtures were investigated at sintering temperatures of 1000, 1100, and 1200 °C. The green ceramic mixtures contained chlorite and dolomite in Tc, and mullite, anorthite, quartz, magnetite, and spinel in FA.

Typical processes that run in this mixture were the liberation of physically bound water, dehydroxylation of talc, transformation of quartz, decomposition of dolomite, transformation of phyllosilicates into spinel phase and mullite, creation of anorthite (at 1000 °C), and sintering.

At 1000 and 1100 °C, the sintering and creation of new phases produced a shrinkage of the sample volume and a decrease in total porosity. Sintering at 1100 °C resulted in a higher shrinkage and the smallest pores. Sintering at 1200 °C caused the moisture expansion due to CaO and the decomposition of talc to amorphous magnesium silicate (enstatite), amorphous silica, and water vapor as well as the crystallization of sapphirine and osumilite, demonstrating incomplete reaction of MgO with the glass phase to the ceramic cordierite.

The experiments carried out have shown promising results from binary mixtures of fly ash without previous treatment to the fly ash–talc-based ceramic bodies. Prepared ceramics at lower sintering temperature due to the K-binding minerals may have similar properties with the commercial ceramics produced at the sintering temperatures of 1250–1300 °C.

**Author Contributions:** Conceptualization, M.V.; Methodology, V.B, A.M., S.Š., and S.V.; Software, J.T.; Validation, M.V., V.B., A.M., and J.T.; Formal analysis, A.M., S.Š., and S.V.; Investigation, M.V.; Data curation, M.V.; Writing—M.V. and J.T.; Writing—review and editing, M.V.; Visualization, M.V. and J.T.; Supervision, M.V.; Project administration, M.V. All authors have read and agreed to the published version of the manuscript.

**Funding:** The work was supported by the ERDF "Institute of Environmental Technology–Excellent Research" (no. CZ.02.1.01/0.0/0.0/16_019/0000853) and the Large Research Infrastructure ENRE-GAT (project no. LM2018098).

**Institutional Review Board Statement:** Not applicable.

**Informed Consent Statement:** Not applicable.

**Data Availability Statement:** The data presented in this study are available on request from the corresponding author.

**Acknowledgments:** The authors would like to thank Jozef Vlcek for providing and reference on the fly ash from power plant, as well as the reviewers and editor for the manuscript revision.

**Conflicts of Interest:** The authors declare no conflict of interest. Authors declare any personal circumstances or interest that may be perceived as inappropriately influencing the representation or interpretation of reported research results. The funders had no role in the design of the study; in the collection, analyses, or interpretation of data; in the writing of the manuscript, or in the decision to publish the results.

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
