# Peer review of "Effect of Talc in Mixtures with Fly Ash on Sintering Crystalline Phases and Porosity of Mullite-Cordierite Ceramics"

_minerals, doi:10.3390/min11020154_

Round 1

Reviewer 1 Report

  1. Poor quality of drawing 9a. The text in the drawing is unreadable.

Please consider the note below in the next articles.

To extend the information on the properties of ceramic materials, strength tests should be carried out. Comparing the strength of the obtained materials (with AF) to classic ceramic products would allow for the extension of information important from the practical point of view. It is also worth to analyze samples based on DIN or EU standards, regarding to raw materials used in the production of ceramics. The bending tests of products  containing AF would be a great supplement of research.

Author Response

  1. Poor quality of drawing 9a. The text in the drawing is unreadable.

Figure 9a was corrected.

Please consider the note below in the next articles.

To extend the information on the properties of ceramic materials, strength tests should be carried out. Comparing the strength of the obtained materials (with AF) to classic ceramic products would allow for the extension of information important from the practical point of view.

It is also worth to analyze samples based on DIN or EU standards, regarding to raw materials used in the production of ceramics. The bending tests of products containing AF would be a great supplement of research.

Authors thank and highly appreciate the valuable comments.

Reviewer 2 Report

Abstract
Line 21/22: «… talc, clinochlore and dolomite transformation at fly ash mullite and cordierite ceramic substrate.” --> Fragment?

Introduction
Please carefully check the language (especially on lines 49-52, and 65). All in all, the introduction could use some structuring.

Results
The resolution of the figures needs to be improved and possibly even the font of the labels in the figures can be increased for better readability. Especially bad are fig. 8 & 9a, where the labels can bearly be read.
Poor formatting of the tables – even though the type setters will probably save this, one gets almost sea sick from table 4.

Line 182: “hallo” --> “halo”
Figure 7a: a scale or other size reference would be helpful
Lines 339 to 353: English!

Conclusions
Again: please check the language carefully.

Author Response

Abstract
Line 21/22: «… talc, clinochlore and dolomite transformation at fly ash mullite and cordierite ceramic substrate.” --> Fragment?

This sentence was rewritten:

The X-ray diffraction, thermal DTA/TGA and mercury intrusion porosimetry analyses were applied to characterize  mineral phases transformation of talc, clinochlore and dolomite and fly ash  in cordierite ceramic.

Introduction
Please carefully check the language (especially on lines 49-52, and 65). All in all, the introduction could use some structuring.

This part was corrected to:

Huffman et al. [6] characterized ash fusion by a number of tests and analytical techniques and indicated temperatures of change to the formation of liquid phase and existence of phases identified on ternary phase diagrams. The partial ash melting occurred at 400 °C and increased at temperatures from 900 to 1200 °C under reducing conditions was controlled by the phases occurring in the FeO-Al2O3-SiO2 phase diagram. Under oxidizing conditions up to 1200 °C, the amount of potassium-bearing minerals in the coal were low-temperature fluxing elements to influenced the amount of the glassy phase.

Results
The resolution of the figures needs to be improved and possibly even the font of the labels in the figures can be increased for better readability. Especially bad are fig. 8 & 9a, where the labels can bearly be read.
Figures 8 and 9a were corrected.

Poor formatting of the tables – even though the type setters will probably save this, one gets almost sea sick from table 4.

Table 4 was corrected.

Line 182: “hallo” --> “halo”

It was corrected

Figure 7a: a scale or other size reference would be helpful

Figure was corrected by adding cm3

Lines 339 to 353: English!

Conclusions
Again: please check the language carefully.

Yes, English was corrected through the text.
